# What Solar–Terrestrial Link Researchers Should Know about Interplanetary Drivers

**Yuri I. Yermolaev** [1,*] ⬤, **Irina G. Lodkina** [1], **Lidia A. Dremukhina** [2], **Michael Y. Yermolaev** [1] and **Alexander A. Khokhlachev** [1]

1 Space Research Institute (IKI RAN), 117997 Moscow, Russia; irina-priem@mail.ru (I.G.L.); michaely2@yandex.ru (M.Y.Y.); aleks.xaa@yandex.ru (A.A.K.)
2 Pushkov Institute of Terrestrial Magnetism, Ionosphere, and Radio Wave Propagation (IZMIRAN), Troitsk, 108840 Moscow, Russia; lidadrem@yandex.ru
* Correspondence: yermol@iki.rssi.ru

**Abstract:** One of the most promising methods of research in solar–terrestrial physics is the comparison of the responses of the magnetosphere–ionosphere–atmosphere system to various types of interplanetary disturbances (so-called "interplanetary drivers"). Numerous studies have shown that different types of drivers result in different reactions of the system for identical variations in the interplanetary magnetic field. In particular, the sheaths—compression regions before fast interplanetary CMEs (ICMEs)—have higher efficiency in terms of the generation of magnetic storms than ICMEs. The growing popularity of this method of research is accompanied by the growth of incorrect methodological approaches in such studies. These errors can be divided into four main classes: (i) using incorrect data with the identification of driver types published in other studies; (ii) using incorrect methods to identify the types of drivers and, as a result, misclassify the causes of magnetospheric-ionospheric disturbances; (iii) ignoring a frequent case with a complex, composite, nature of the driver (the presence of a sequence of several simple drivers) and matching the system response with only one of the drivers; for example, a magnetic storm is often generated by a sheath in front of ICME, although the authors consider these events to be a so-called "CME-induced" storm, rather than a "sheath-induced" storm; (iv) ignoring the compression regions before the fast CME in the case when there is no interplanetary shock (IS) in front of the compression region ("sheath without IS" or the so-called "lost driver"), although this type of driver generates about 10% of moderate and large magnetic storms. Possible ways of solving this problem are discussed.

**Keywords:** solar wind; interplanetary drivers; solar–terrestrial physics

## 1. Introduction

Pioneering studies in the 1960s and 1970s [1–5] showed that disturbances in the magnetosphere are mainly associated with the appearance of the southward ($Bz < 0$) component of the interplanetary magnetic field (IMF). The IMF lies in the ecliptic plane under steady interplanetary conditions, and a substantial $Bz < 0$ component is observed only in disturbed types of solar wind (SW), such as corotating interaction regions (CIRs) between slow and fast SW streams, interplanetary coronal mass ejections (ICMEs), and sheath compression regions in front of fast ICMEs (see the reviews by [6–8]). All drivers have increased IMF; the compression regions (CIR and sheath) have higher values of density, temperature, and β-parameter (the ratio of thermal pressure to magnetic pressure) than undisturbed SW, and the ICME has lower values of these parameters than SW. Magnetic clouds are often distinguished from ICME, which have a higher and more regular IMF than another subclass of ICME, ejecta. Many studies have shown different magnetospheric responses to various types SW, even for close values of IMF $Bz$ (see, e.g., [9–39] and references therein). This approach seems very promising, since it allows for the discovery of new physical connections in solar–terrestrial physics. There is currently a steady upward trend in the

number of studies in which some physical processes in the magnetosphere, ionosphere, and atmosphere are compared with some specific types of SW. However, the results of such studies are often questioned on the basis of inappropriate methodological approaches, small event sample sizes, and absence of statistical significance tests. The main reason for this is that most researchers on the solar–terrestrial link are not specialists in SW phenomena and make mistakes in identifying interplanetary drivers, often leading to incorrect conclusions. The most common errors are associated with the use of incorrect criteria to identify the types of SW, either by the authors of the erroneous work or by the authors of data sources that are used by other researchers. Typical examples of such methodical errors were considered in detail in our previous studies [40–42] and are not considered in depth in this article.

In this paper, we consider two other, physical classes of incorrect approaches that lead to erroneous conclusions about the relationship between interplanetary drivers and magnetospheric disturbances. In the first of these approaches, the authors suggest that the disturbance of the magnetosphere–ionosphere system is caused by a "CME-induced" phenomenon and does not take into account the fact that a CME in the solar corona can result in a sequence of two single drivers, a sheath compression region and an ICME (interplanetary CME including ejecta or magnetic cloud (MC)). As has been shown [22,24,25,34–37], the sheath has higher efficiency in terms of the generation of magnetic storms than an ICME; although magnetic storms are often induced by a sheath, authors tend to consider the ICME to be the cause of disturbance, rather than the sheath. Secondly, the solar wind can contain types of disturbances that we call "lost drivers". Very often, the compression region sheath before the ICME is not considered by some authors as a driver, if there is no interplanetary shock (IS) in front of the sheath, while this type of driver, "sheath without IS", can generate moderate and strong magnetic storms with Dst $< -50$ nT [31].

This paper is structured as follows: Section 2 describes the data and methods used, Section 3 presents the results of the measurements and their analyses, and Section 4 discusses and summarizes the results.

## 2. Data and Methods

Our investigation is based on the 1 h OMNI data of interplanetary plasma and magnetic field measurements and magnetospheric indices (http://omniweb.gsfc.nasa.gov (accessed on 10 January 2021), [43]). Unlike our previous works, this paper contains data for an extended interval of 45 years (1976–2020).

To identify the corresponding large-scale types of SW, we use the threshold criteria for the key parameters of SW and IMF for each 1 h point in the archive (see the paper [44], and the site with web addresses ftp://ftp.iki.rssi.ru/pub/omni/ or http://www.iki.rssi.ru/pub/omni). Our method for the identification of SW types is based on criteria that are similar to those described in many previous papers (see reviews [12,45,46] and references therein); our identification results agree with those of other authors (e.g., [47–50]; however, unlike in other similar studies which identified only selected types of SW, we use a general set of threshold criteria for all types of SW and carry out an identification for each 1 h point.

To analyze the magnetospheric response to changes in interplanetary conditions, we selected the following disturbed types of SW: two types of ICMEs (MC and ejecta), two types of sheaths (sheath before MC, $SH_{MC}$, and sheath before ejecta, $SH_{EJ}$), the corotating interaction region (CIR), and the forward ISs in front of three types of compression region $SH_{MC}$, $SH_{EJ}$, and CIR. MC differs from ejecta mainly in its higher and more regular magnetic field. These and other differences are described in more detail in [30,40,51]. As already shown earlier (see, for example, [30]), the difference in the properties of $SH_{MC}$ and $SH_{EJ}$ is small (i.e., their properties depend little on the type of ICME piston—MC or ejecta), but their properties depend on the presence of shock ISs (i.e., they depend on the speed of propagation of ICMEs relative to the preceding quiet solar wind). In this paper, we examine these differences using data spanning a longer time interval than in previous works.

We use the double superposed epoch analysis (DSEA) method with two reference time instants at the ends of the interval [26]. Similar methods were used for analysis of MC profiles in [47,48] and for the statistic study of magnetic storms [52]. In this method, all intervals of a certain type of SW are divided into an equal number of equal subintervals, regardless of the actual duration of each interval, and the parameters are averaged in these subintervals. This procedure allows one to compute the average time profile of any parameter for events with different durations.

We assume that a magnetic storm is generated by a driver if the minimum of the Dst index is fixed in the interval of this driver or within 2 h of its end. A period of 2 h corresponds to the average time delay between the peak in the southward IMF $Bz$ component and the associated Dst peak of an intense magnetic storm [16,53,54].

## 3. Results

### 3.1. Distribution of Dst Index in Sheath + ICME Complexes

In our papers [30,40] using the double superposed epoch analysis method, we studied the average behavior of interplanetary and magnetospheric parameters for the eight most common sequences of SW phenomena: (1) SW/ejecta/SW, (2) SW/sheath/ejecta/SW, (3) SW/IS/sheath/ejecta/SW, (4) SW/MC/SW, (5) SW/sheath/MC/SW, (6) SW/IS/sheath/MC/SW, (7) SW/CIR/SW, and (8) SW/IS/CIR/SW (here, SW means undisturbed solar wind) for the period 1976–2000, and we showed that the average temporal profiles of the magnetospheric indices have maxima in the intervals from the last part of the sheath to the beginning part of the ICME. In particular, the panels in the first and third rows of Figure 1 present the average temporal profiles of the measured Dst. These results agree well with those previously published for the shorter time interval 1976–2000 (panels "e" of Figures 3–8 in [30]).These profiles are divided into two parts: (1) the drop in the Dst and Dst* indices that is observed in the sheath and their minima in the first hours of MC and ejecta, and (2) the moderate growth in the Dst and Dst* indices in the MC and the ejecta.

Unlike previous works [30,40], in this article, we present the results of a similar analysis performed over an extended time interval 1976–2020 and show the profiles of individual events in addition to the average ones (see panels in first and third rows of Figure 1). The results are not qualitatively different from previous results; however, doubling the number of events increased the reliability of the results. The time distributions in the sheath and ejecta/MC intervals for (1) the onset of storms with Dst $<-50$ nT (blue columns) and (2) the Dst minimum (red columns) are shown by panels in the second and fourth rows of Figure 1. It should be noted that the blue and red columns were calculated using five equal time subintervals in sheaths or ejecta/MC, although, in the figure, they are shifted with respect to each other for clarity. These data confirm that a large number of magnetic storms began at the beginning of sheath, and the maximal number of Dst index minima (the maxima of magnetic storms) was recorded in the intervals from the last part of the sheath to the beginning part of the ejecta/MC.

### 3.2. Temporal Profiles of Parameters in Sheath + ICME Complexes

Table 1 and Figures 2–5 allow us to compare the sheath characteristics for four variants of the SW phenomena sequences, namely, sheath/ejecta, IS/sheath/ejecta, sheath/MC, and IS/sheath/MC. In contrast to Section 3.1, where all the sheath events were analyzed, here, we analyze only those sheath events that generated magnetic storms with Dst $<-50$ nT. The table shows the number of sheath events in two sets: the total number of registered events in each of the four identified subspecies, K1, and the number of K2 events from K1 for which complete sets of measurements were recorded, including sheath and ICME intervals. K1 events are used to estimate the occurrence rate of sheath events, and K2 events are used to calculate the average values of the parameters in the table and average time profiles in Figures 2–5. K2 events are shown in brackets in the table.

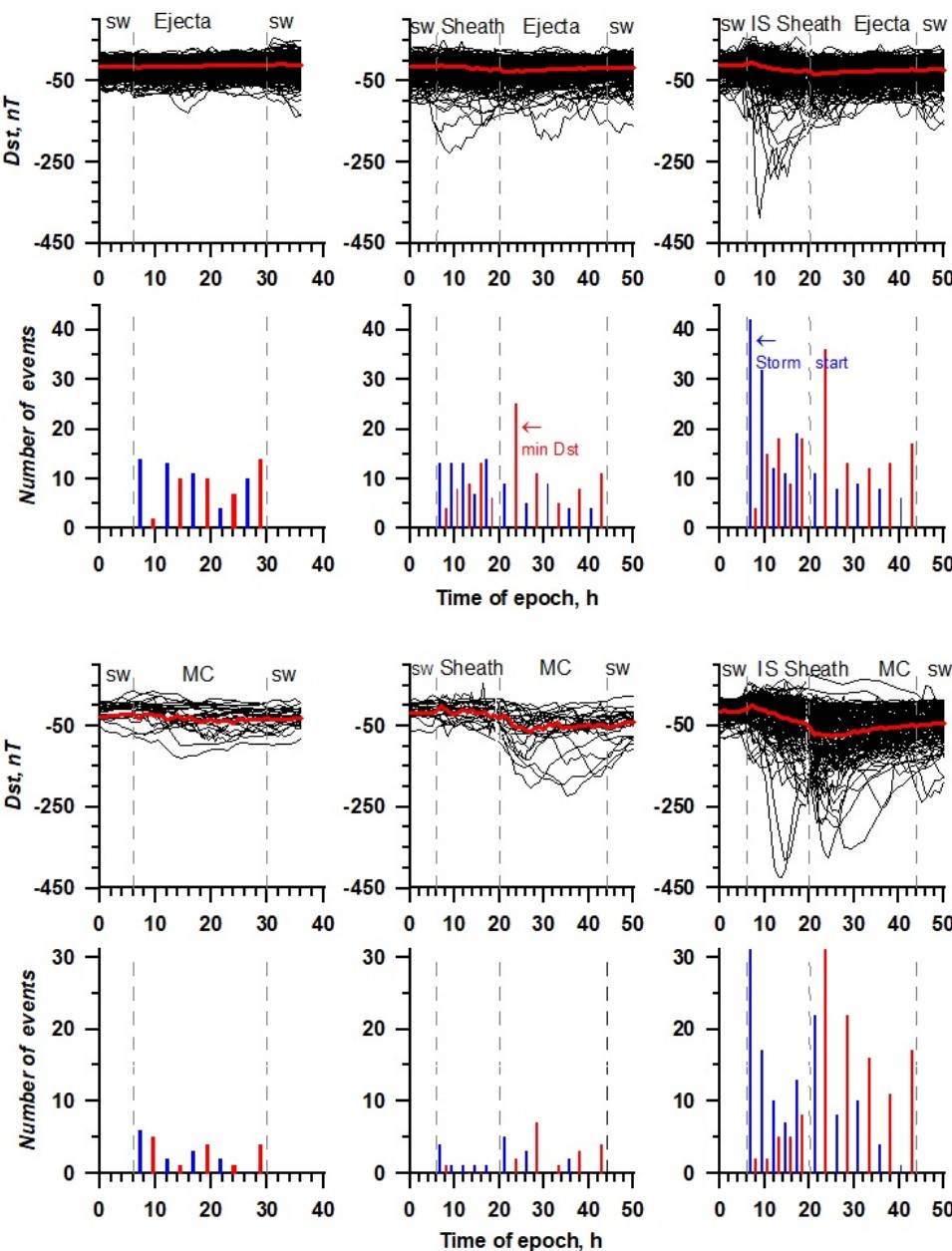

**Figure 1.** The panels in first and third rows show the temporal profiles of Dst index for individual phenomena (black lines) and average profile of Dst index for them (red line) in six different sequences of solar wind phenomena. The panels in the second and fourth rows show the distributions in the sheath or ejecta/MC time intervals, the number of onsets of storms (blue columns), and the number of maxima (Dst index minima) of storms (red columns). Vertical dashed lines indicate (from right to left) (1) the last point of the ejecta/MC intervals, (2) the first point of the ejecta/MC intervals, and (3) (in the presence of a sheath region) the first point of the sheath intervals.

The number of sheath phenomena in front of ejecta without IS (439) was slightly higher than the number of phenomena with IS (395), while the number of sheath phenomena in front of the MCs with IS (160) was significantly higher than the number of phenomena without IS (28). As in many similar estimates, the average values for many parameters of Table 1 turned out to be large and close in magnitude to the standard deviations. However, the statistical error (i.e., the standard deviation divided by the square root of the number of measurement points) for some of them turned out to be small, and, in this case, the differences in the average values for different types of sheath can be considered statistically

significant [55]. In particular, the data for durations of sheaths with IS before ejecta and MC (third and fifth columns in Table 1) allow one to estimate statistical errors for both types as about 0.5 h and allow suggesting that the mean duration of sheath phenomena in front of the ejecta was longer than for sheath phenomena in front of the MC. The number of magnetic storms generated by the sheath phenomena in front of the ejecta, with and without IS, was almost the same (63 and 59). For the MC phenomena, the difference was higher (25 and 3), but it is necessary to note that this result was obtained with a small number of events with the MC compared with the ejecta.

**Table 1.** Mean values and standard deviations of sheath parameters for four SW sequences.

|  | Sheath/Ejecta | IS/Sheath/Ejecta | Sheath/MC | IS/Sheath/MC |
|---|---|---|---|---|
| Number of K1 events | 439 | 395 | 28 | 160 |
| (Number of K2 events) | (329) | (360) | (24) | (155) |
| Duration of events, h | $14.0 \pm 8.8$ | $16.3 \pm 9.5$ | $13.1 \pm 9.8$ | $12.1 \pm 6.1$ |
| Number of magnetic storms | 59 | 63 | 3 | 25 |
| $V$, km/s | $439 \pm 95$ | $459 \pm 107$ | $432 \pm 91$ | $491 \pm 141$ |
| $T$ ($10^5$), K | $1.57 \pm 1.35$ | $1.77 \pm 1.73$ | $1.55 \pm 1.52$ | $2.44 \pm 3.63$ |
| $T/Texp$ | $2.07 \pm 1.06$ | $2.0 \pm 1.1$ | $1.99 \pm 1.09$ | $2.08 \pm 1.61$ |
| $N$, cm$^{-3}$ | $9.6 \pm 6.4$ | $12.4 \pm 9.4$ | $13.4 \pm 8.4$ | $16.1 \pm 11.1$ |
| $B$, nT | $8.1 \pm 3.6$ | $9.9 \pm 4.7$ | $9.8 \pm 5.1$ | $13.2 \pm 7.7$ |
| $Kp*10$ | $29 \pm 15$ | $33 \pm 15$ | $32 \pm 16$ | $42 \pm 19$ |
| $Dst$, nT | $-17 \pm 27$ | $-19 \pm 36$ | $-18 \pm 27$ | $-24 \pm 54$ |
| $Dst*$, nT | $-22 \pm 29$ | $-28 \pm 38$ | $-27 \pm 27$ | $-37 \pm 52$ |
| $AE$, nT | $276 \pm 249$ | $327 \pm 285$ | $319 \pm 317$ | $449 \pm 391$ |

Figures 2–5 show the average time profiles of interplanetary parameters and magnetospheric indices for four SW sequences including sheath phenomena: (1) SW/sheath/ejecta, (2) SW/IS/sheath/ejecta, (3) SW/sheath/MC, and (4) SW/IS/sheath/MC. The figures include the sheath region processed with the method of double superposed epoch analysis (DSEA, points from 6–19) and regions of SW and ICME (including MC or ejecta) processed with the simple method of superposed epoch analysis (SEA, points from 0–5 and from 20–25, respectively). The 10 panels of each figure show the average time profiles of the following parameters:

(a)  the thermal pressure Pt, the ratio of the thermal pressure and magnetic one β, and the relative density of α-particles Na/Np;

(b)  the proton temperature T × 10$^{-5}$ K, and the ratio of the measured temperature and temperature estimated on the basis of average velocity-temperature relation T/Texp [44–46,50];

(c)  the longitude and latitude angles of the bulk velocity vector phi and theta;

(d)  the components of the electric field Ey and the IMF *Bz*;

(e)  the measured Dst and density corrected Dst* indices (in contrast to Dst, the Dst* index is cleared of the contribution of the current at the magnetopause and is mainly determined by the ring current);

(f)  the dynamic pressure Pd, and the magnitude of the IMF B;

(g)  the components of the IMF Bx and IMF By;

(h)  the Alfvenic Va and sound Vs speeds;

(i)  the ion density N and Kp index;

(j)  the proton bulk velocity V, and the AE index.

For "sheath with IS" events followed by ejecta, the values of parameters such as the thermal and dynamic pressure Pt and Pd, the magnitude of the magnetic field B, the proton temperature T and T/Texp, the bulk plasma velocity V, and the Dst and Dst* indices

are larger than for the "sheath without IS" events (Figures 2 and 3). This behavior of parameters is connected with a sharper increase in these parameters if the sheath region begins with IS. After a period of 2–3 h after the start of the sheath region, these parameters change in a similar way. The situation is similar for the "sheath with IS" and "sheath without IS" events followed by MC (Figures 4 and 5). For sheath events with subsequent MC, the values of the parameters Pt, Pd, B, T, T/Texp, V, Dst, and Dst* are higher than when there is a subsequent ejecta event.

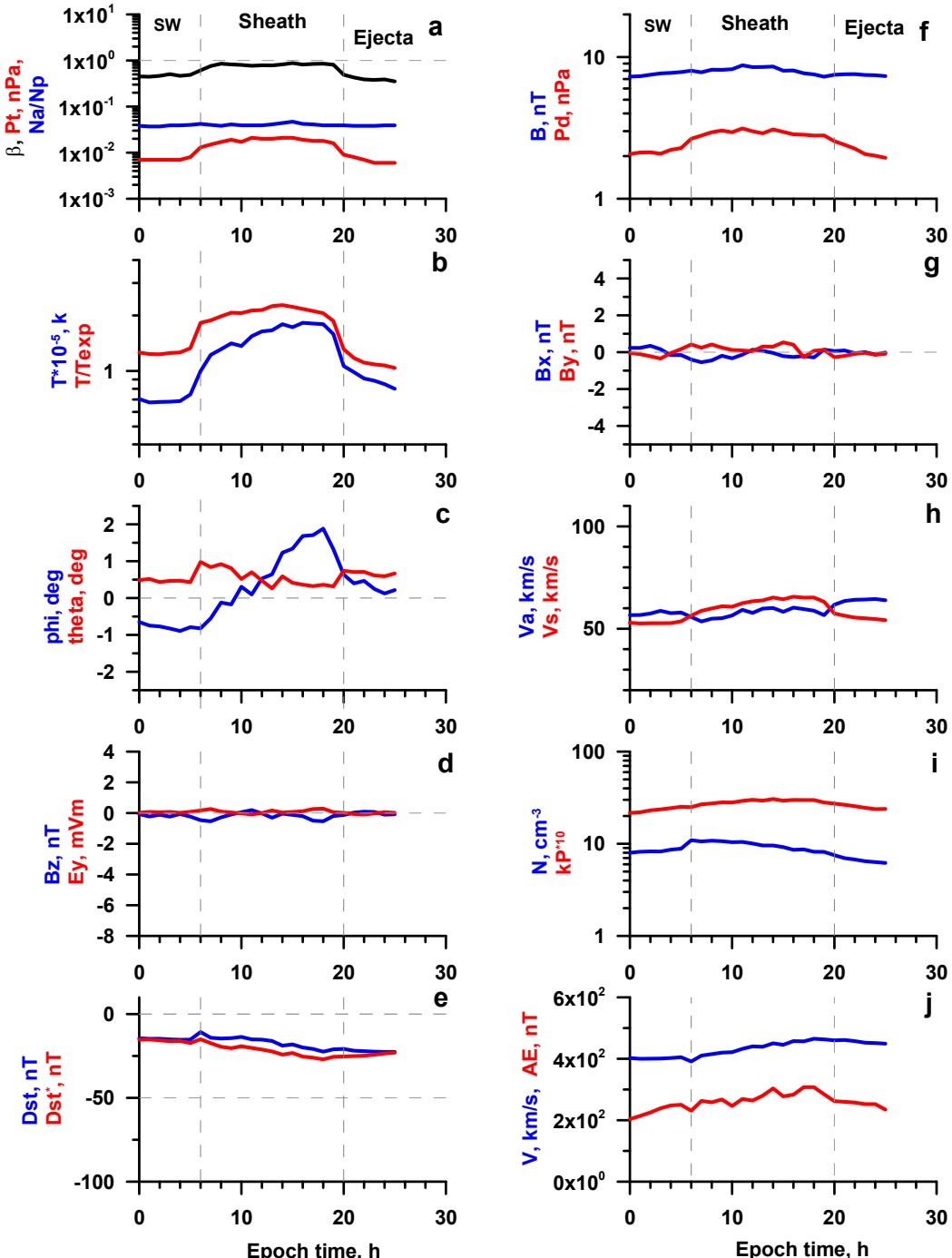

**Figure 2.** The temporal profiles of the solar wind parameters and magnetospheric indices for the sheath/ejecta sequence obtained using the SEA method (without rescaling) for SW (points from 0–5) and for ejecta (points from 20–25) and using the DSEA method (with rescaling) for sheath (points from 6–19).

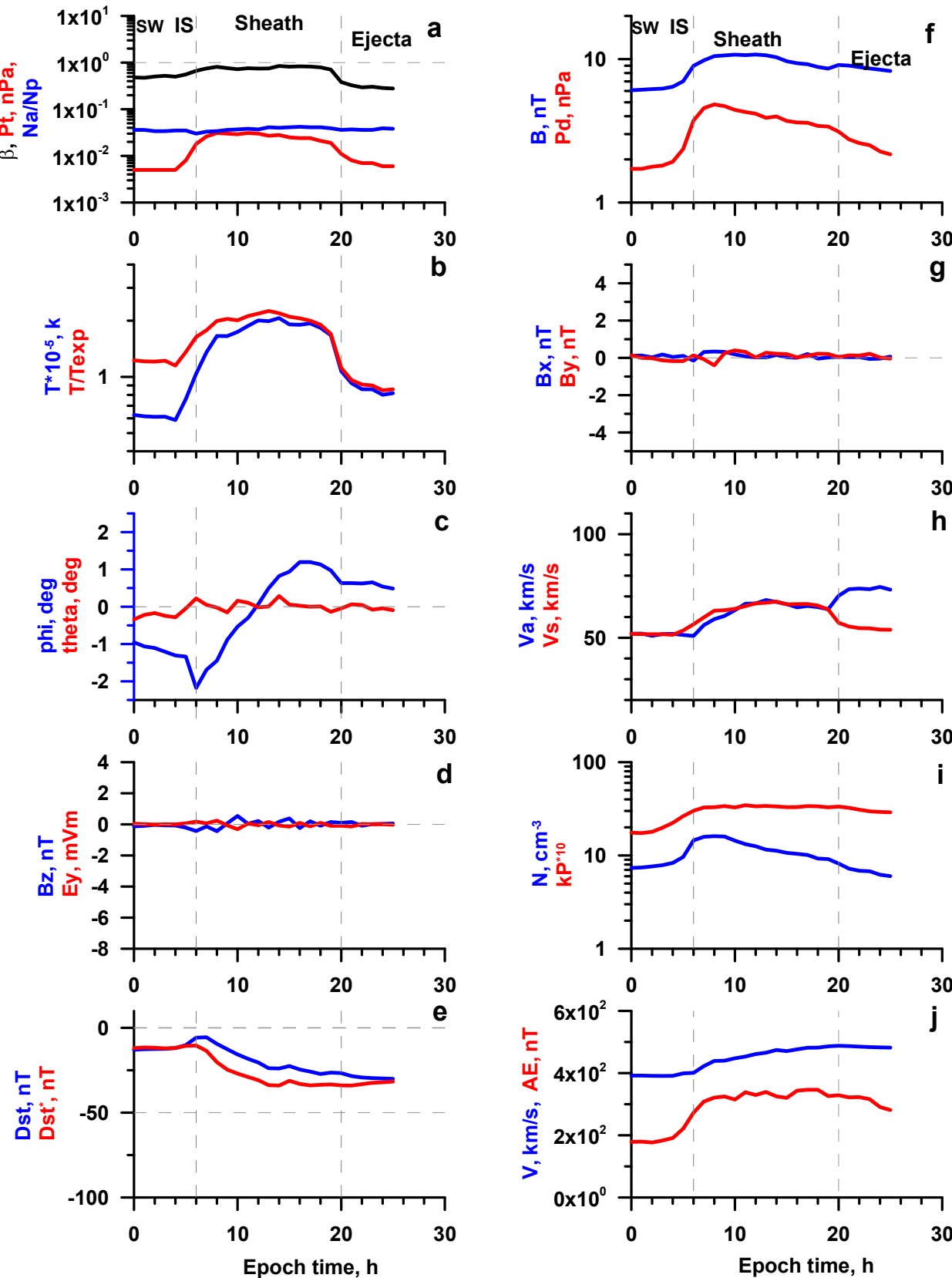

**Figure 3.** As in Figure 2 for the IS/sheath/ejecta sequence.

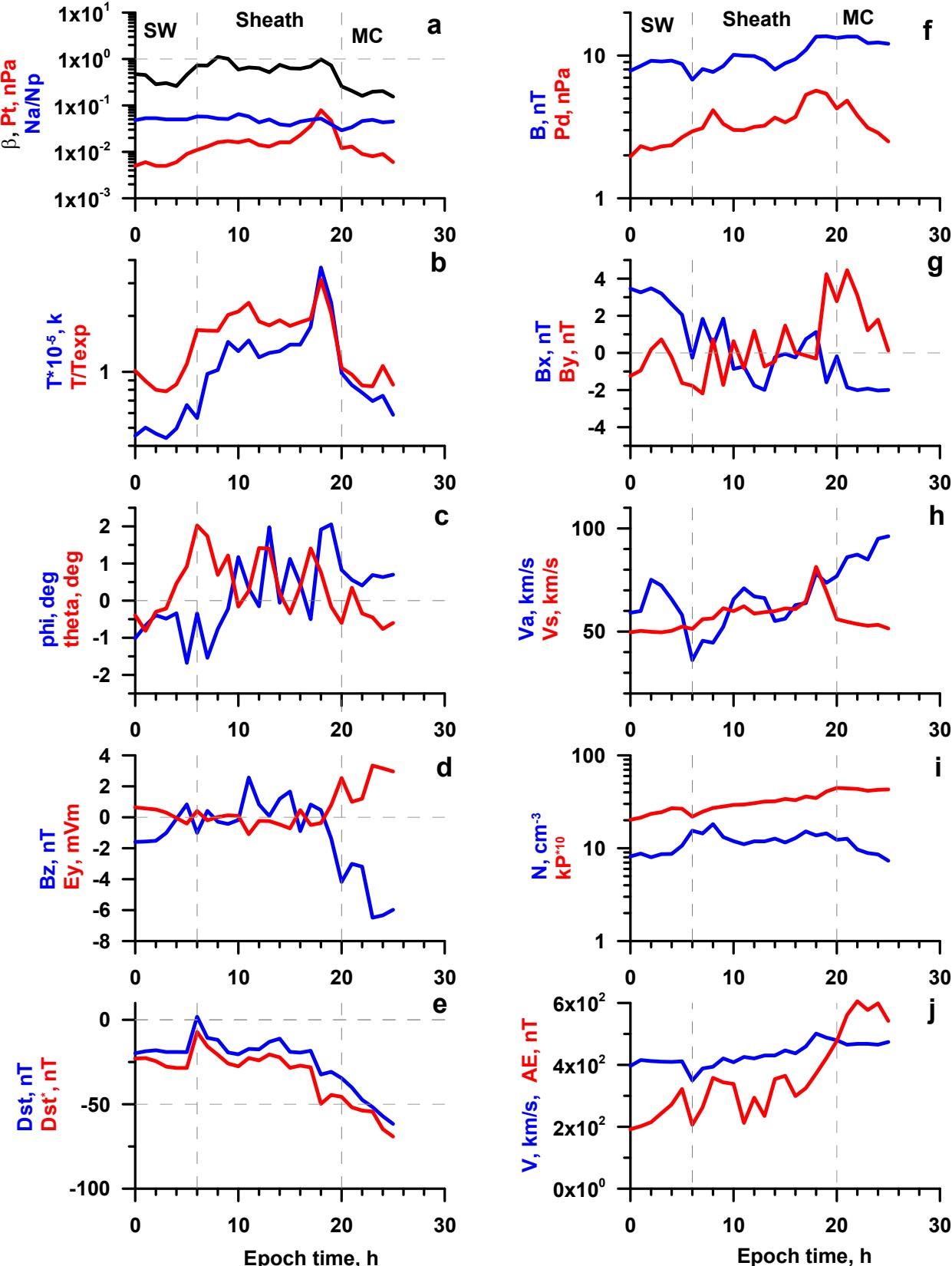

**Figure 4.** As in Figure 2 for the sheath/MC sequence.

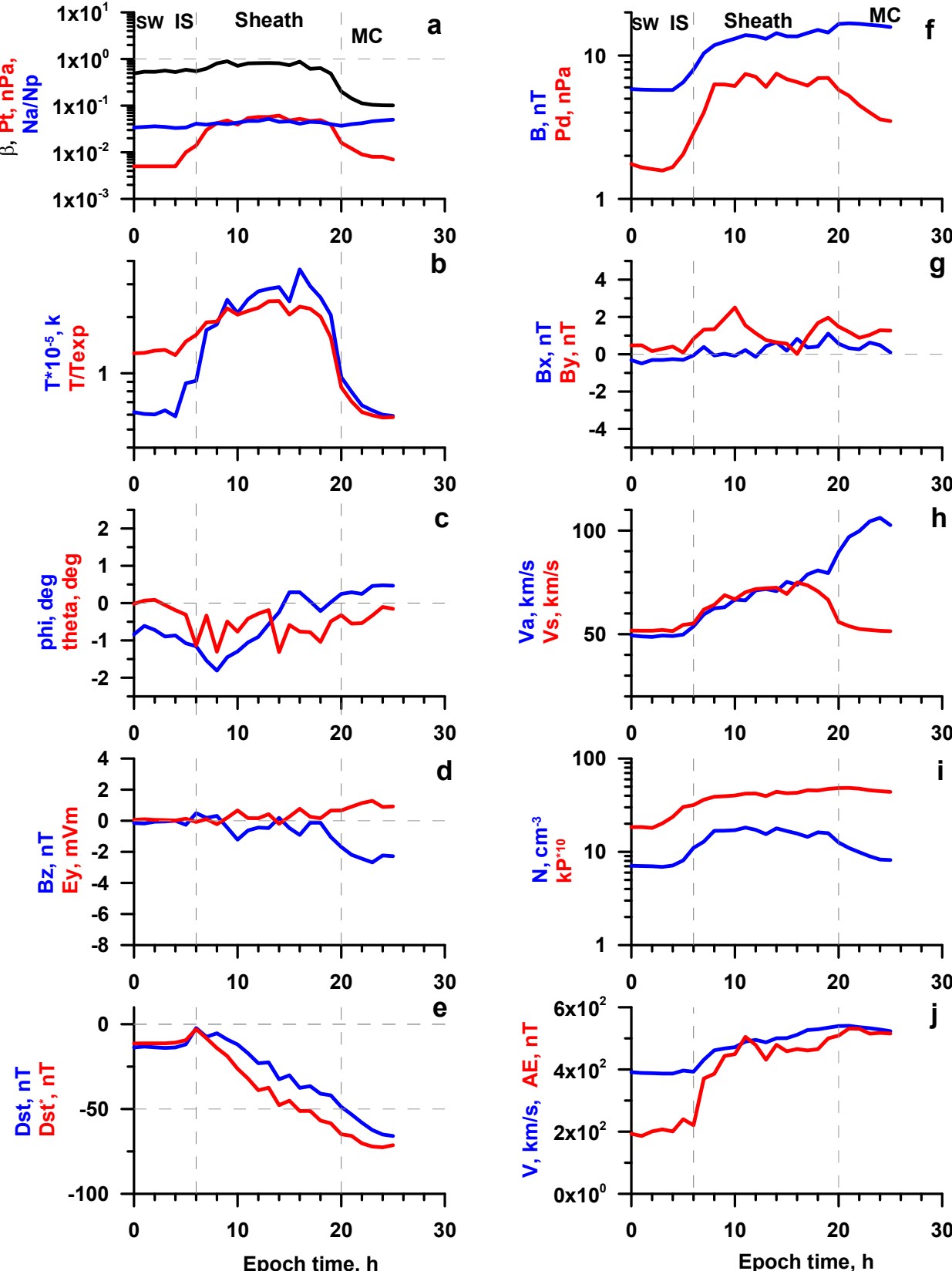

**Figure 5.** As in Figure 2 for the IS/sheath/MC sequence.

## 4. Discussion and Conclusions

In this work, we presented the distribution of parameters and Dst and Dst* indices for complex interplanetary drivers including sheath and ICME during the period 1976–2020, and we summarized the incorrect approaches most frequently used by authors when associating magnetosphere–ionosphere–atmosphere disturbances with various interplanetary drivers. The most common mistake is the misidentification of the interplanetary drivers for this association; although we did not consider this type of error in detail in this paper, it must be emphasised that this is responsible for the highest number of papers with incorrect conclusions. There are two main reasons for this: the authors use incorrect criteria to identify types of drivers or use sources with incorrect driver identification (in particular, previously published articles). An analysis of works containing this type of error shows that some articles may turn out to be "toxic", i.e., they serve as a source of error for several further articles that use the incorrect driver identification in these publications as source data in their analysis [40–42]. To reduce the number of errors of this kind, we believe it is necessary to impose more stringent requirements when reviewing articles that use data on the identification of interplanetary drivers.

The authors of many works studied so-called "CME-induced" storms (or other types of magnetospheric disturbance) as a special type of storm. In our opinion, there are no CME-induced disturbances, although there are sheath-induced and MC/ejecta-induced disturbances, as well as multistep disturbances, which are excited by a sequence of sheath/MC or sheath/ejecta events. The data presented here confirm that these "CME-induced" disturbances of the magnetosphere are in fact responses to completely different interplanetary drivers or the successive impact of different drivers [40]. The sheath and ICME (including MC and ejecta) have different physical origins and different properties, and they may have different mechanisms for generating disturbances in the inner parts of the magnetosphere.

Compression regions in front of the ICME are often not considered as sheath, if they do not have IS in the beginning of region, and they are not analyzed as the cause of the disturbance of the magnetosphere–ionosphere–atmospheric system. The "sheath without shock" region, which is recorded in front of an ICME almost as often as a sheath region with IS, is quite geoeffective and is the source of about 10% of moderate and large storms with Dst $<-50$ nT [31].

Several experimental facts should be mentioned. Firstly, the sheath events have an average value of IMF B higher than that of ejecta and an average value close to that of MC events [30]. Secondly, sheath events have a magnetic storm generation efficiency ~50% higher than that for ICME (including MCs and ejecta) [22,25,27,34–36]; that is, with the same IMF southward components, sheath events generate magnetic storms ~1.5 times stronger than ICME. Therefore, we can conclude that the approach, where the contribution of sheath compression regions (including the lost driver of "sheath without shock") to the generation of storms is not taken into account, is incorrect, and the sheath role is often underestimated. This erroneous approach often results in incorrect conclusions being drawn when studying solar–terrestrial links.

We would like to point out that the general paradigm has changed in recent years, and more and more researchers are using more detailed information to study the impact of interplanetary drivers on the inner regions of the Earth's magnetosphere. We hope that our article will draw the attention of the scientific community to the serious problem of the significant increase in the number of publications with incorrect identification of interplanetary drivers of magnetospheric–ionospheric–atmospheric disturbances. These publications contain incorrect conclusions and discredit the progressive approach to the study of solar–terrestrial relations. We believe that one of the possible ways to solve this problem could be the creation of an agreed catalog of interplanetary drivers by an international group of experts and the use of this catalog in problems of solar–terrestrial physics.

**Author Contributions:** Conceptualization, Y.I.Y.; data curation, I.G.L.; formal analysis, I.G.L. and A.A.K.; investigation, L.A.D. and M.Y.Y.; methodology, Y.I.Y. and I.G.L.; software, I.G.L. and A.A.K.; visualization, I.G.L. and M.Y.Y.; writing—original draft, Y.I.Y. and L.A.D.; writing—review and editing, Y.I.Y., I.G.L., and M.Y.Y. All authors have read and agreed to the published version of the manuscript.

**Funding:** This research was funded by the Russian Foundation of Basic Research, grant number 19-02-00177a.

**Data Availability Statement:** The authors are grateful to the developers of the OMNI database (http://omniweb.gsfc.nasa.gov (accessed on 10 January 2021)). Data on the identification of large-scale types of solar wind for 1976–2020 are available from the site of the Space Research Institute, Moscow, Russia, with web addresses ftp://ftp.iki.rssi.ru/pub/omni/ or http://www.iki.rssi.ru/pub/omni (accessed on 10 January 2021).

**Acknowledgments:** Yuri Yermolaev thanks SCOSTEP's "Variability of the Sun and Its Terrestrial Impact" (VarSITI) program for supporting his participation in the Closing Symposium, 10–14 June 2019, Sofia, Bulgaria.

**Conflicts of Interest:** The authors declare no conflict of interest.

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
