# Peer review of "What Solar–Terrestrial Link Researchers Should Know about Interplanetary Drivers"

_universe, doi:10.3390/universe7050138_

Round 1

Reviewer 1 Report

The work is complete enough and does not require any major modifications or additions. My only complaint is that the authors reference their previous publications a bit too much, while they could easily spend a few extra sentences to clarify several parts of their methodology. Since some readers might not be familiar with their past work, and especially since some points that they make pertain to very recent publications (e.g. Yermolaev et al., 2020) I would advise them to add just a few sentences to further elucidate some key points: 

Lines 66-69: It is hard for me to picture an ICME without a shock front. Does this mean that the discontinuities that are typically associated with shocks are too small to be identified? Could you please add a few extra sentences to describe this part more thoroughly, especially since the referenced publications is very recent and I do not have access to it.  

Lines 99-105: I think it would be beneficial to the paper if the authors add a few lines here to explain the DSEA. At the very least, just mention the two time instants that are critical for the method (onset and Dst minimum) and elaborate on the rescaling/averaging that is done in-between these two. Also, please explain what happens for the measurements that are made outside this interval. 

Lines 120-125:  I think that referencing a figure from another paper is more confusing than helpful at this point. I would advise you to limit the description in this paragraph to only the relevant panels of Fig.1 of this work. 

Lines 184-194: Please define at least some of the parameters here that are not obvious to everyone, i.e. Texp and Dst* (if the latter has not been defined previously). 

Discussion and Conclusions: To end the paper in a more positive note, I would advise to note that the general paradigm in recent years has indeed changed and more and more researchers use more detailed criteria for the identification of SW drivers. (two example works that come to mind are the following, but you are free to choose whichever you want [I am not part of any of these publications, nor had I any contribution to them]) 

Kilpua, E. K. J., H. Hietala, D. L. Turner, H. E. J. Koskinen, T. I. Pulkkinen, J. V. Rodriguez, G. D. Reeves, S. G. Claudepierre, and H. E. Spence (2015), Unraveling the drivers of the storm time radiation belt response, Geophys. Res. Lett., 42, doi:10.1002/2015GL063542. 

Kilpua, E. K. J., Fontaine, D., Moissard, C., Ala-Lahti, M., Palmerio, E., Yordanova, E. et al. (2019). Solar wind properties and geospace impact of coronal mass ejection-driven sheath regions: Variation and driver dependence. Space Weather, 17, 1257–1280. https://doi.org/10.1029/2019SW002217 

A final comment: The authors mention that magnetic storms could very well emerge due to multi-step disturbances, which are excited by a sequence of events. Another aspect that plays an important role is the preconditioning of the magnetosphere due to previous storms that might create favorable conditions for the excitation of even more energetic phenomena. Is there any way to account for this in your future works? e.g by including some sort of parameter for the “recent disturbance level” 

Editorial Notes:  

Please, make sure the fonts are consistent throughout the paper. The name of the last author in the title page is in boldface, and the font sizes in lines 208-218 are sometimes smaller and sometimes larger than the rest of the manuscript.  

Sections “Author Contributions”, “Funding”, “Data Availability Statement” and “Conflicts of interest” are not properly prepared! Also, in the “Acknowledgements” section, a name is missing and has been replaced by the characters “YY”.  Please address these issues.

Reviewer 2 Report

This manuscript reports an extension of work already published and covers an additional 20 years of data from OMNI. My main concern is about the statistical analysis of the data. My overall comments are as follows.

The abstract exposes the errors of other authors, but does not present how the work described in the article contributes to resolving these errors. Please add an explanation.

Line 77: I think the authors mean 1976-2020. There are many gaps in OMNI data pre-1995. The authors should comment on how they take account of this, and whether there may be risk of affecting their results. The authors could check their results further by using data only over the period 1995 - 2020. Having read further, may this be the reason for using only the K2 number of events? Please clarify.

Lines 165-167: This is very unclear: The standard deviation is the square root of the variance, and the variance is equal to (sum of the squared differences from the mean) divided by the number of measurements. Please show the correct standard deviations in the table so that we can see which parameters varied and which are within the errors. Only by doing this the differences in behaviour for the four cases considered can be firmly established.

Lines 169-171: The mean durations are all consistent within the errors in the Table.

Line 185: ‘expected’ on the basis of what?

Lines 208-218: Since the plots show no errors it is very difficult to conclude where there are real difference among the four cases. The results reported would be convincing if the Table was to show statistically significant differences among the four cases.

Lines 225-227: How can the authors conclude that misidentification of the interplanetary drivers is responsible for incorrect conclusions without having considered this type of error in detail? The authors should clarify and strengthen the statistical significance of their results in order to support their discussion statements and demonstrate convincingly where other authors are going wrong.

Editorial suggestions

Line 49: for this is that most researchers

Line 84: unlike in other similar studies which identified

Line 90: in front of three types of compression

Line 105: profile of any parameter

Line 133: It should be noted

Table: normally would be ‘Table 1’

Lines 154-155: the total number K1 of registered events in each of the four identified subspecies and the number K2 of events for which complete sets

Line 164: Should this be 28?

Round 2

Reviewer 2 Report

The authors have taken into account some of my comments. I am still concerned that the standard deviations in the Table are so large that parameters such as the durations of events are all consistent with each other (and the authors admit this and agree with my comment on lines 169-171, which they define as unclear). So how can it be statistically justified that "In particular, the data in the Table show that the mean duration of sheath phenomena in front of the ejecta was longer than for sheath phenomena in front of the MC."?

What is "the statistical error (i.e. the standard deviation divided by the square root of the number of measurement points)"? Isn't the standard deviation already divided by the square root of the number of measurements? Is this a problem of definitions? If the 'errors' are smaller, why not to show them in the Table?

Here I am not concerned with the reasons for the differences, I am concerned with demonstrating that the differences are real.

If these points were explained the paper would be acceptable for publication, in my opinion.
